# Drivers of Fire Anomalies in the Brazilian Amazon: Lessons Learned from the 2019 Fire Crisis

**Marcus V. F. Silveira** [1,*]**, Caio A. Petri** [1,†]**, Igor S. Broggio** [2,†]**, Gabriel O. Chagas** [1]**, Mateus S. Macul** [1]**, Cândida C. S. S. Leite** [1]**, Edson M. M. Ferrari** [1]**, Carolina G. V. Amim** [3]**, Ana L. R. Freitas** [1]**, Alline Z. V. Motta** [4]**, Luiza M. E. Carvalho** [4]**, Celso H. L. Silva Junior** [1]**, Liana O. Anderson** [5] **and Luiz E. O. C. Aragão** [1,6]

1  Earth Observation and Geoinformatics Division, National Institute for Space Research, São José dos Campos 12227-010, Brazil; caio.petri@inpe.br (C.A.P.); gabriel.chagas@inpe.br (G.O.C.); mateus.macul@inpe.br (M.S.M.); candida.leite@inpe.br (C.C.S.S.L.); edson.ferrari@inpe.br (E.M.M.F.); ana.defreitas@inpe.br (A.L.R.F.); celso.junior@inpe.br (C.H.L.S.J.); luiz.aragao@inpe.br (L.E.O.C.A.)
2  Laboratory of Environmental Sciences, Centro de Biociências e Biotecnologia, Universidade Estadual do Norte Fluminense Darcy Ribeiro (UENF), Campos dos Goytacazes 28013-600, Brazil; igor@pq.uenf.br
3  Institute of Science and Technology, São Paulo State University (UNESP)/National Centre for Monitoring and Early Warning of Natural Disasters (CEMADEN), São José dos Campos 12247-004, Brazil; carolina.vergeti@unesp.br
4  Department of Forestry, Universidade Federal dos Vales do Jequitinhonha e Mucuri (UFVJM), Diamantina 39100-000, Brazil; alline.zagnoli@ufvjm.edu.br (A.Z.V.M.); luiza.esteves@ufvjm.edu.br (L.M.E.C.)
5  National Centre for Monitoring and Early Warning of Natural Disasters (CEMADEN), São Jose dos Campos 12227-010, Brazil; liana.anderson@cemaden.gov.br
6  College of Life and Environmental Sciences Amory Building, University of Exeter, Rennes Drive, Exeter EX4 4RJ, UK
*  Correspondence: markgvenf@gmail.com
†  These authors contributed equally to this work.

**Abstract:** The 2019 fire crisis in Amazonia dominated global news and triggered fundamental questions about the possible causes behind it. Here we performed an in-depth investigation of the drivers of active fire anomalies in the Brazilian Amazon biome. We assessed a 2003–2019 time-series of active fires, deforestation, and water deficit and evaluated potential drivers of active fire occurrence in 2019, at the biome-scale, state level, and local level. Our results revealed abnormally high monthly fire counts in 2019 for the states of Acre, Amazonas, and Roraima. These states also differed from others by exhibiting in this year extreme levels of deforestation. Areas in 2019 with active fire occurrence significantly greater than the average across the biome had, on average, three times more active fires in the three previous years, six times more deforestation in 2019, and five times more deforestation in the five previous years. Approximately one-third of yearly active fires from 2003 to 2019 occurred up to 1 km from deforested areas in the same year, and one-third of deforested areas in a given year were located up to 500 m from deforested areas in the previous year. These findings provide critical information to support strategic decisions for fire prevention policies and fire combat actions.

**Keywords:** land use change; forest fires; deforestation; water deficit; wildfires; fire activity; land tenure; MODIS MCD14DL; MapBiomas; fragmentation

## 1. Introduction

The Amazon has been the biome in Brazil with the highest number of active fires over the last two decades, having on average almost two times more active fires than the Cerrado biome [1]. Ignitions in the Amazon are in fact most entirely caused by the extensive processes of forest conversion and agricultural maintenance [2]. Fires are used to remove the remaining biomass following deforestation, a process known as slash-and-burn. In this process, ashes help to fertilize the soil in preparation for the new agricultural use [3,4]. After the consolidation of agricultural areas, farmers may continue to use fire during the dry season for vegetation renewal and weed control. Because of the low adoption of fire management practices in the region [3], both fires associated to deforestation or agricultural practices can occasionally escape to adjacent fields and forests and cause wildfires [5].

Forest fires in the Amazon can lead to a multitude of negative impacts linked to the positive feedback between land use change and climate exacerbating fire activity. As fragmentation increases with deforestation, the microclimate along forest edges becomes drier, leading to increased tree mortality rates, which in turn increases fuel loads, turning forests into a fire-prone system [6,7]. Fire intensity also increases in forests previously affected by fires due to changes in microclimate and fuel loading [8]. Coupled with these effects, forest flammability is highly aggravated during drought years [9], which predominantly occur in the Amazon as a consequence of negative rainfall anomalies driven by the El Niño and the Atlantic Multidecadal Oscillation [10]. After fire disturbances, forests experience changes in structure and species composition [11], in addition to a slow biomass recovery that can last for decades and probably centuries [12]. Gross $CO_2$ emissions from forest fires in the Amazon, when coupled with deforestation emissions, can offset the positive net carbon sink from old-growth and secondary forests in the region, contributing to the exacerbation of global climate change [13]. In years of extreme drought, $CO_2$ emissions from forest fires can even surpass those from deforestation [14]. The negative impacts of an intensive fire activity, in general, also extend to the socioeconomic sphere. Uncontrolled fires can bring financial losses to farmers by destroying crops, fences and house infrastructure [15–17], while the number of hospitalizations for respiratory ailments in the Amazon increase during fire season [18,19], which raises the government's expenditure with health treatments.

Global observational and modeling studies have showed different fire occurrence patterns in space and time for different ecosystems as a function of human and climatic drivers [20–22]. Human drivers for instance are positively associated with burned area in tropical forests [20] and outweigh climatic drivers in South America [21]. In such regions, a comprehensive understanding of which and where anthropic factors are most related to fire activity is therefore important as it has direct implications for implementing more effective fire prevention policies and for allocating fire brigades. Such knowledge is, however, still poorly developed for real-world applications. Brazil's National System for Forest Fire Prevention and Control (Prevfogo), for example, plans the allocation of fire brigades by assessing the municipalities with the highest fire counts in previous years, but this procedure leads to an accuracy rate of only 30%, as observed by Morello et al. [23].

In 2019, fire occurrence in August peaked across the Brazilian Amazon. The event grabbed media's attention and incited protests around the globe, raising discussions on the role played by the increasing deforestation over the last years [24,25]. One remarkable peculiarity of the 2019 fire season occurred on August 10, which became known as the *Fire Day*. Investigations by the police point that farmers and merchants coordinated a joint light of fire in agricultural and deforested areas during this day, but the actual contribution of the Fire Day for 2019 fire season remains obscure, so does the real gravity of the 2019 fire crisis in comparison to other years. Under not only national but also international pressure, Brazil's government convoked an operation (Decree n° 9985/19) [26] referred as the *Guarantee of Law and Order* (GLO), which authorized the use of army forces to tackle the fires during the period spanning from August 24 to October 24 of 2019. This was the first time in the country's history that military forces were used for leading fire and deforestation combat in the Amazon.

Although the total number of active fires in the Brazilian Amazon in 2019 had a similar magnitude to those of 2016 and 2018 [1], this study is centered on the hypothesis that abnormally high fire occurrence across space, months, states and land tenure can be hidden in general-biome comparisons of yearly fire counts, and thus need to be identified in order to be properly addressed in a context suitable for decision-making. In this study, hence, we aim to perform a comprehensive analysis of active fire anomalies and their drivers in the Brazilian Amazon Biome to support strategic plans for fire prevention and fire suppression. Fire anomalies in 2019 were assessed across time and space and compared with patterns from the 2003–2018 time-series. The assessment of drivers was focused on the widely known case of Amazonian fire crisis in 2019, considering that drivers in this year are likely to continue exerting their influence over the next years with similar magnitudes. We explored a wide range of potential drivers for fire occurrence, quantifying their contribution and assessing their relationship with fire across different scales, from the biome to states and finally to a local level.

## 2. Materials and Methods

### 2.1. Datasets and Data Preprocessing

#### 2.1.1. Active Fires

Daily active fire data from 2003 to 2019 were acquired from MODIS MCD14DL product, which is the only dataset available over this timespan. The product is provided as point vectors in the Fire Information for Resource Management System (FIRMS) platform [27] and each point represents the centroid of a 1 km by 1 km pixel in which one or more fires were detected by the algorithm, from both Aqua and Terra satellites. For this study, we combined active fire data from Aqua daytime and Terra nighttime, which retrieved the highest number of detections while presenting lower redundancy as compared to data collection from both satellites in daytime and nighttime [28]. We removed from the final active fire dataset all active fires with a confidence level equal to or lower than 30% [29]. As shown by Giglio et al. [30], the MCD14DL algorithm has a robust performance against false detections, with rates around 4% for South America overall.

#### 2.1.2. Deforestation of Old-growth Forests

Spatial data for annual deforestation of old-growth forests from 2003 to 2019 were collected from PRODES Amazonia [31], which is Brazil's official deforestation monitoring program for the Amazon. PRODES maps deforested areas (clear-cut) based on mid-resolution satellite images, primarily from the Landsat program (30 m spatial resolution), but also from Sentinel 2 (10–20 m) and CBERS 4 (10–20 m) satellites. The PRODES information for a given year (reference year) correspond to deforestation events that occurred approximately from the period between August 1st of the previous year to July 31th of the reference year. The PRODES Amazonia program is restricted to areas under the natural domain of a forest formation and only old-growth forest deforestation is accounted for, excluding secondary forests dynamics.

#### 2.1.3. Maximum Cumulative Water Deficit

We used the Maximum Cumulative Water Deficit (MCWD) as a measure of water stress in forests. This metric was proposed by Aragão et al. [32] and is based on the idea that the average monthly evapotranspiration (E) in Amazonian forests is around 100 mm, as reported by several studies in different regions of the biome [33–35]. Thus, a monthly precipitation below 100 mm will result in a greater loss of water by E in relation to the rainfall input, therefore generating a water deficit state. This metric does not account for different soil conditions such as water storage capacity, and plant physiological adaptations such as stomatal control. To generate the MCWD, we first collected daily rainfall data from the Climate Hazards Group InfraRed Precipitation with Station (CHIRPS) product, which provides rainfall estimates based on satellite information, physiographic indicators,

and interpolations from in situ weather stations, with 0.05° spatial resolution [36]. Costa et al. [37] compared rainfall estimations from CHIRPS with field gauge stations and found a $R^2$ superior to 0.9 for the regions encompassing the Brazilian Legal Amazon, while Anderson et al. [38] found a $R^2$ of 0.73 for the Amazon biome, with a root mean square error below 15 mm per month. We proceeded by subtracting 100 mm from monthly rainfall values to generate the monthly Cumulative Water Deficit (CWD) for each pixel in CHIRPS dataset, as in the following formula:

$$\text{If } WD_{n-1}(i,j) - E(i,j) + P_n(i,j) < 0$$
$$\text{then } WD_n(i,j) = WD_{n-1}(i,j) - E(i,j) + P_n(i,j); \quad (1)$$
$$\text{else } WD_n(i,j) = 0$$

The calculation started with the year of 2003 and proceeded until 2019. The MCWD of each pixel for a given year is considered the most negative (minimum) monthly CWD value. Because the MCWD metric does not have positive values, the lowest the MCWD (more negative values), the highest the water deficit.

### 2.1.4. Drivers of Fire Occurrence in 2019

We gathered 14 potential drivers that could have directly or indirectly influenced fire activity in 2019 (Table 1). In addition to deforestation and MCWD in 2019, we collected data from previous active fires (2016–2018), previous deforestation (2014–2018), pastureland area, cropland area, three age classes of secondary forests (1–5, 6–15, and 16–33 years old), three fragmentation metrics (total forest edge, mean fragment area, and number of fragments), the nearest distance to federal highways, and rural population density. Land-use and land-cover data for 2018 were derived from MapBiomas [39], which provides a comprehensive annual classification of Brazil's lands from 1985 onwards. Maps from MapBiomas have a 30 m spatial resolution and are based on satellite images from the Landsat program. Secondary forest ages in 2018 were derived from the dataset produced by Silva Junior et al. [40], which was based on the MapBiomas time series, with a starting point in 1986. From MapBiomas' forest classification of 2018, which makes no distinction between old-growth and secondary forests, we generated metrics of fragmentation with the R package *landscapemetrics* [41].

**Table 1.** List of potential drivers of fire occurrence in 2019, in the Amazon, evaluated by this study.

| Variable | Units | Description | Data Source |
|---|---|---|---|
| Previous fire | Fire count | Total active fire count in the three previous years (i.e., 2018, 2017, and 2016). Each fire count represents one or more active fires within a 1 × 1 km pixel | MODIS MCD14DL |
| Deforestation in 2019 | $km^2$ | Old-growth deforestation from around August 2018 to July 2019 | PRODES/INPE |
| Previous deforestation | $km^2$ | Old-growth deforestation in the five previous years (i.e., from 2014 to 2018) | PRODES/INPE |
| Pasture area | $km^2$ | Area in 2018 | Mapbiomas |
| Cropland area | $km^2$ | Area in 2018 | Mapbiomas |
| Forest Age I | $km^2$ | Secondary forest area 1–5 years old in 2018 | Silva Junior et al. [40] |
| Forest Age 2 | $km^2$ | Secondary forest area 6–15 years old in 2018 | Silva Junior et al. [40] |
| Forest Age 3 | $km^2$ | Secondary forest area 16–33 years old in 2018 | Silva Junior et al. [40] |
| Total forest edge | km | Total length of forest (including old-growth and secondary) edge | Mapbiomas |
| Mean fragment area | $km^2$ | Mean forest (including old-growth and secondary) fragment area | Mapbiomas |
| Number of fragments | Patch count | Total number of forest (including old-growth and secondary) fragments | Mapbiomas |
| Maximum Cumulative Water Deficit | mm | Minimum value of the monthly accumulated water deficit in 2019 | CHIRPS |
| Distance to roads | km | Euclidean distance to the nearest federal highway | DNIT |
| Rural population density | Population count | Total number of inhabitants in rural areas per grid cell (100 $km^2$). Data were collected from Brazil's 2010 Population Census | IBGE |

Note: DNIT stands for National Department of Transport Infrastructure, IBGE stands for Brazilian Institute of Geography and Statistics.

2.1.5. Data Aggregation into Grid Cells

For the assessments involving annual active fire counts, deforestation and MCWD, and for our driver inspection analyses, we stratified the whole biome into 10 × 10 km grid cells to extract statistics based on a sampling scheme, while accounting for spatial variations. Data for all variables were aggregated into grid cells with a sum function, with the exception of MCWD for which we retained the minimum value from pixels intersecting a cell, and the nearest distance to roads that was directly calculated from grid cells.

*2.2. Time-Series Evaluation and Identification of Monthly Fire Anomalies*

We started the analysis assessing the extent of droughts in our time-series to understand if the climatic scenario of the year 2019 was favorable for the fire crisis. For each year from 2003 to 2019, we calculated the proportion of grid cells with MCWD values within each of these five ranges: 0 mm, −1 to −200 mm, −200 to −300 mm, −300 to −400 mm, and <−400 mm.

Next, we quantified the total monthly active fire counts in 2019, comparing them with the average of drought years and non-drought years of the time-series, and the maximum and minimum range of non-drought years. Years with extreme drought events refer to 2005, 2010, 2015, and 2016, as reported by Aragão et al. [14] and Aragão et al. [42]. We defined a monthly active fire occurrence in 2019 as anomalous if it was similar or greater than the average of drought years, and/or out of the range of non-drought years. This quantification was performed for the biome in general and for the states encompassing it. Amazonian states were also compared with respect to annual of active fire counts, deforestation, and mean MCWD. The state of Tocantins was not considered in the analyses given the very small portion of its territory falling within the biome.

*2.3. Spatial Distribution of Annual Fire Anomalies*

We employed a spatially explicit approach to identify anomalies in annual active fire counts, using the grid cells as samples. We defined two possible ways in which grid cells could be considered anomalous in 2019: across-grid anomaly and maximum fire count.

Across-grid anomaly: This was the major type of anomaly assessed throughout this study and refer to cells with a z-score for active fire occurrence at least two standard deviations above the average across the biome, for a given year. Thus, we selected only cells with significant positive fire anomalies at the 0.05 level. We transformed cell values into z-scores for each year of our time-series and quantified the number of years each cell in our grid was considered anomalous. This procedure allowed us to identify the average recurrence of anomalous cells in 2019, and to identify anomalous cells for the first time in 2019.

Maximum fire count: cells in which the total number of fires in 2019 was the maximum of the time-series. This metric represents a form of temporal anomaly that does not depend upon cross-comparisons with other cells as in the across-grid anomaly. Cells with fire detections for the first time within the time-series were also represented here.

We calculated the distribution of both these anomaly types across the Amazonian states and across six land tenure categories: private lands, agrarian settlements, undesignated public lands, indigenous lands, protected areas of strict use, and protected areas of sustainable use. Data for protected areas were extracted from Brazil's Ministry of the Environment (MMA) Geoportal [43], regularized indigenous lands were extracted from *Fundação Nacional do Índio* (Funai) portal [44], and the remaining land tenures were derived from Imaflora's platform [45].

Additionally, we assessed a time-series of annual active fires, deforestation and mean MCWD in cells with across-grid anomalies in 2019, and quantified their monthly share of burned area in 2019. Monthly burned area data were derived from MODIS and have a spatial resolution of 500 m. More information about this product algorithm can be accessed at Giglio et al. [46].

Lastly, we verified the relationship between the number of years in which cells were across-grid anomalous and the total agricultural area in a cell's space, assuming that areas with a higher recurrence have been more deforested and occupied with agricultural areas. The proportion of agricultural area was calculated over the total area of agriculture (pasture and croplands) plus natural vegetation cover (forest, savanna, shrubland, and grassland formations, excluding mangroves) in each cell, according to Mapbiomas' classification in 2018. All other land cover/land use classes were removed because of their lack of relationship with fire occurrence and to enable cross-comparisons, since the amount of area able to be occupied by agriculture varies among cells (some cells for instance are half covered with water bodies).

*2.4. Inspection of Drivers for Active Fire Occurrence in 2019*

Initially, we performed a visual assessment of active fire occurrence in 2019 across different ranges of magnitude in each driver's distribution of values. Because of the large number of grid cells (43,074), we retained in this analysis only those that had at least one active fire in 2019. We stratified grid cells on a quantile basis into five intervals of similar sizes for each driver to guarantee a representative comparison across intervals, and then removed outliers to facilitate the visualization. The subsequent sections will describe the procedure to evaluate the importance of drivers at the biome level, state level and local level.

2.4.1. Assessment of Drivers at Biome and State Levels

Our goal in this approach was to test the capacity of each driver to discriminate between cells with across-grid fire anomalies cells and cells with non-anomalous fire occurrence in 2019, for an understanding of the possible factors leading to an aggravation of fire occurrence.

For this purpose, we used the filterVarImp function in R's caret package. For each driver, this algorithm computes the two class probabilities across a continuum of cutoffs in the driver distribution, then generates a Receiver Operating Characteristic (ROC) curve, and calculates the area under the curve (AUC) [47]. The *y*-axis in a ROC curve plot represents the proportion of events correctly predicted (i.e., true positive rate), and the *x*-axis represents the proportion of nonevents incorrectly classified as events (i.e., false-positive rate). Events or positives for this study correspond to the across-grid anomaly class. The area under the curve (AUC) ranges from 0 to 1 and serves as a measure of feature importance, as an AUC of 0.5 would indicate that a variable is completely unable to discriminate between the two classes because the false positive rate increases in the same proportion of the true positive rate. An AUC of 1 on the other hand indicates a variable that is able to completely separate the true classes. In addition to the AUC metric, we also used Welch's t-test to verify the hypothesis that the mean of each driver was either greater or lower in across-grid anomalous cells compared to non-anomalous cells, at the 5% significance level.

2.4.2. Assessment of Drivers at the Local Level

With this approach we aimed to identify regions in the biome where each driver is best related to fire occurrence in 2019. We employed the *local bivariate relationships* analysis from ArcgisPro as a rigorous approach to effectively select only the drivers with significant influence at the local level, and only the places where these drivers have strong relationships with fire occurrence.

We ran the local bivariate analysis for each driver using the grid cells as input samples. The algorithm first estimates the joint entropy of the driver and fire occurrence on a moving-window for the actual data and for a user-specified number of permutations in the variables distribution. Next, the algorithm calculates a *p*-value indicating the probability of obtaining a joint entropy value in the permutations as small as that of the actual data. The lower the entropy of a variable, the more predictable the variable is. The joint entropy is estimated by the sum of individual entropies from the two variables, minus their mutual information. In significant local relationships, the actual data are therefore expected to have a much smaller joint entropy than permutations because the variables share a high level of mutual information (i.e., a state of dependency). The algorithm then applies a False Discovery Rate correction to *p*-values based on a given confidence level to determine which *p*-values

are significant while accounting for the multiple testing problem. For the local relationships defined as significant, the algorithm analyses linear and polynomial regressions on the moving-window in order to classify the relationship across five different types (positive linear, negative linear, concave, convex, and undefined complex). More information about the algorithm can be found in Guo [48].

We set the number of neighbors composing the window size as 40 cells (4000 km$^2$), and defined the number of permutations as 199. As a final step to reduce data dimensionality and facilitate interpretations, we retained only samples with a statistically significant relationship at the 5% significance level, with a positive or negative linear relationship with fire occurrence, and with a R$^2$ equal or greater than 0.5. For all but one variable we retained only positive linear relationships because negative linear relationships were very scarce. Only for the mean fragment area we did the opposite. The way to interpret maps from this analysis is that information in each cell is actually representing the result of a 4000 km$^2$ area that encompasses the analyzed cell and its 40 closest neighboring cells.

*2.5. Proximity Analyses*

In this approach we did not aggregate data into grid cells, but assessed proximity patterns from active fires and deforestation polygons as a solution to provide information for strategy planning of fire prevention and fire combat.

For each year of our time-series, we calculated the proportion of daily active fires within the following distances intervals from deforestation polygons of that same year: 0–0.25 km, 0.25–0.5 km, 0.5–1 km, 1–5 km, and >5 km. We set 250 m as the limit for the first interval, considering that active fire points are the centroid of a 1 × 1 km pixel; thus, fires that were closer to the pixel center were more likely to be detected by the algorithm and more likely to have occurred within or at the border of deforestation polygons. We also assessed distance relationships between deforestation polygons of a given year and those of previous years. More specifically, for each year of our time-series we calculated the proportion of deforestation polygons in that year that were connected, up to 0.5 km, and up to 5 km from deforestation polygons of the respective previous year. In addition, we quantified the proportion of deforestation polygons in a given year connected to deforestation polygons at any previous year of our time-series.

## 3. Results

*3.1. The Extent of Water Deficit in Amazonia from 2003 to 2019*

Extreme drought years differed most remarkably from their previous years by a decrease of around 5–15% (215,300–646,110 km$^2$) in the proportion of cells in the lowest water deficit interval (MCWD from −1 to −200 mm, Figure 1A). In subsequent years, this decrease is recovered, but the proportion of cells with a MCWD lower than −400 mm, which is the most intense water deficit interval, increases around 7–9%. Increases in this last MCWD interval took three years following the 2005 drought, and one year after the 2010 and 2015–2016 droughts.

The year of 2019 was peculiar for having both a decrease in the proportion of cells in the lowest water deficit interval (MCWD greater than −200 mm), similarly to drought years, but also an increase in the proportion of cells in the highest water deficit interval (MCWD lower than −400 mm), similarly to years following extreme drought events. The proportion of cells with a MCWD lower than −200 mm in 2019 was the second greatest of the time-series, only behind 2017. Translating to numbers, this means that around 55% of the biome had MCWD values lower than −200 mm in 2019. This proportion was 69% in 2017, and the average proportion for non-drought years is 47%. Even more remarkably, the proportion of cells with MCWD values lower than −400 mm in 2019 (13.5%) was the highest of the time-series. Water deficit levels in 2019 across the biome usually followed a decreasing gradient from east to west (Figure 1B).

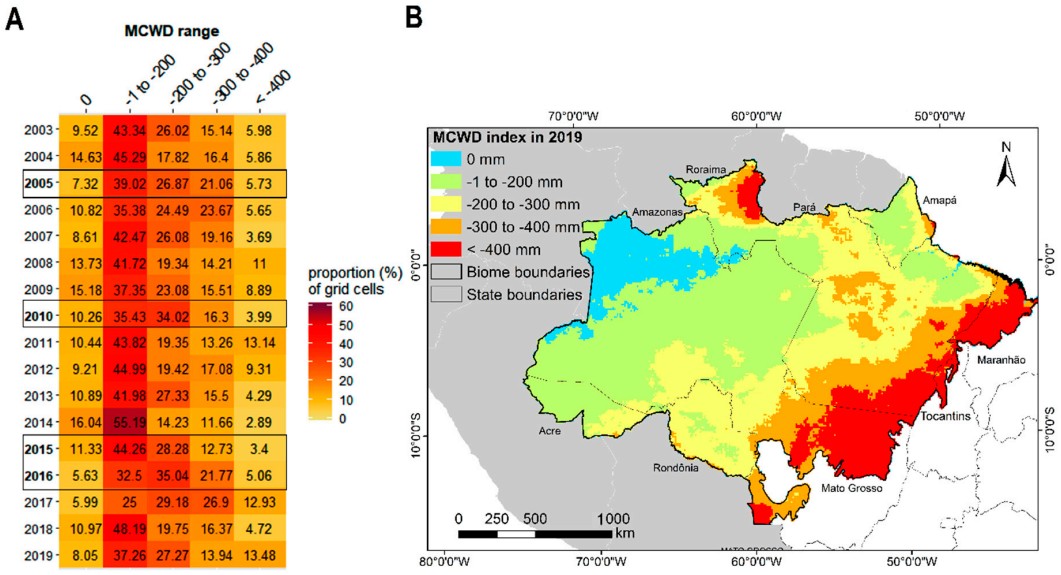

**Figure 1.** (**A**) For each year, the proportion of grid cells with Maximum Cumulative Water Deficit (MCWD) values within each of the five intervals. Years with extreme droughts in the Amazon are highlighted in boxes. (**B**) The spatial distribution of MCWD values in 2019 across five different intervals.

### 3.2. Daily and Monthly Fire Occurrence Dynamics in 2019

Although active fire occurrence in the Fire Day (August 10) was higher than the average of the first ten days of August 2019, it was lower than previous dates such as August 4 and August 5 (Figure 2A). However, the next day following Fire Day had a twofold increase in fire occurrence and reached the second highest level of the year. The peak of the year occurred just four days following Fire Day, with 2909 active fires on August 14. The two weeks that followed the Fire Day until the start of the Guarantee of Law and Order (GLO) on August 24 concentrated one-fifth of all fire occurrences for the entire year. After the start of GLO, fire occurrence in September normalized to a magnitude similar to that observed in August before the Fire Day and decreased in October until the end of the GLO in October 24. Less than one week after the end of GLO, fire occurrence began to increase more than twofold from the average in October and kept similar magnitudes in November.

Most months of 2019 exhibited fire occurrences within the range and near the average of non-drought years, except for March, August, September, and October (Figure 2B). Fire occurrence in March 2019 was greater than any regular year and greater than the average of drought years, while fire occurrence August was greater than the average of regular years, though still below the average of drought years. Fire occurrence in September 2019 was far below the average of regular years, opposing the historical pattern of a peak in fire activity in this month. October 2019 had the lowest fire occurrence across the time-series analyzed.

Although in most states fire activity was centered within the August–September period, linked to the rainfall seasonality in most of Amazonia region, the fire season in Roraima, however, was centered in February–April, and October–November in Amapá and Maranhão. Pará historically tends to have an outburst in fire activity from October to November, which can surpass the annual fire occurrence of some states (Figure 2C).

The states that presented the highest contribution to fire activity in the Amazon have consistently been Mato Grosso (MT), Para (PA), and Rondônia (RO) over the years, with more than 65% of active fires occurring in these regions. Monthly fire occurrence in these states during 2019 was mostly below the average of non-drought years. Abnormally high fire occurrences in 2019 were present in fact in secondary states in terms of fire activity, namely Amazonas (AM), Acre (AC), and Roraima (RR). Fire occurrence in AM during August 2019 was the highest of the time-series. For AC, fire occurrence in August 2019 was also higher than any non-drought year, but close to the average of drought years.

RR has had a very small historical contribution to fire total occurrence in the biome, but fire occurrence in March and April of 2019 were fivefold apart from the averages, being the highest of the time-series.

One-third of fire occurrence in August 2019 occurred in PA (Figure 2D), and the remaining was mostly concentrated in MT, RO, and AM, in similar magnitudes (18%, 20%, and 21%, respectively). RR contributed most entirely to the anomalous magnitude in March 2019 (86%), followed by a small contribution from MT. This last was the state with the steadiest contributions in the first semester of 2019, while contributions from PA predominated in the second semester.

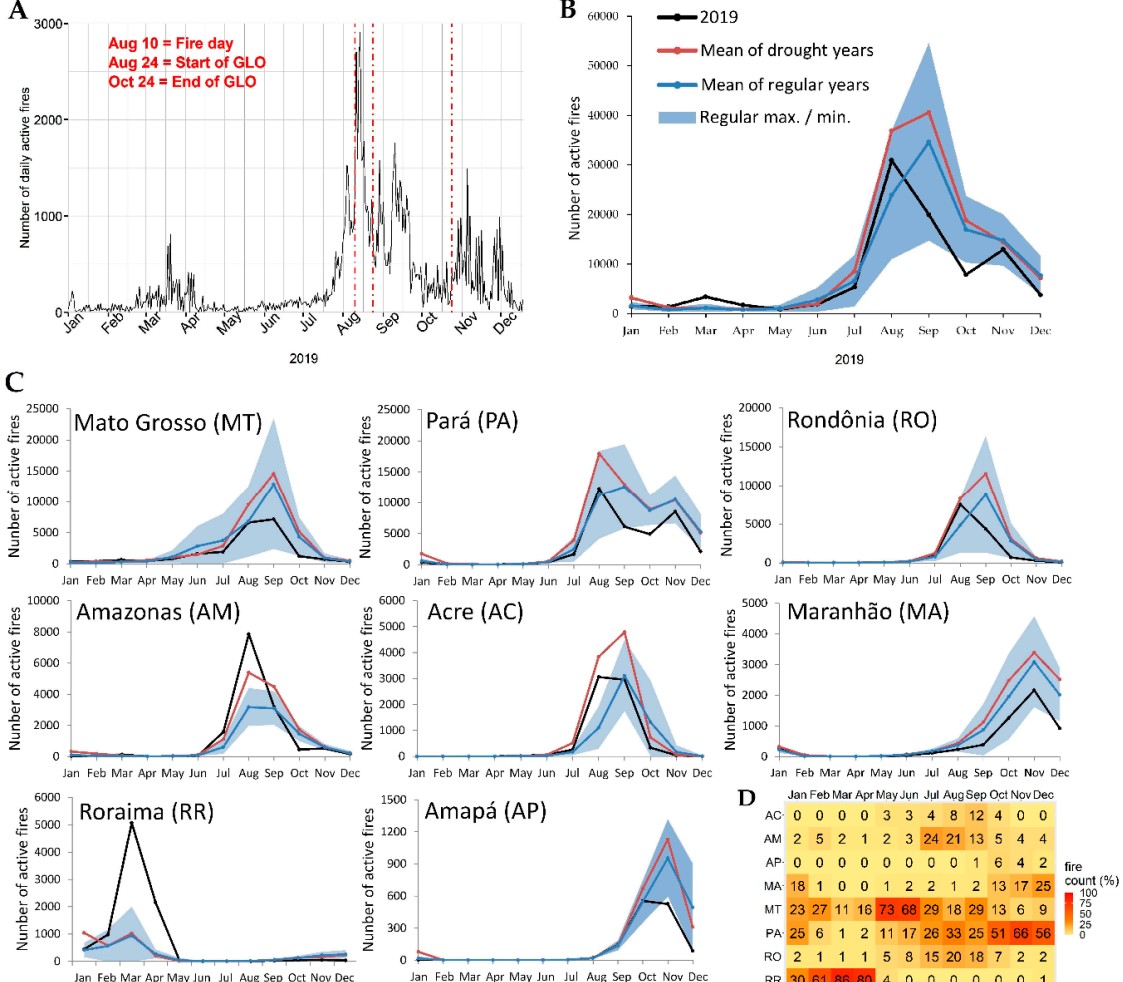

**Figure 2.** (**A**) Daily fire occurrence in 2019, with important dates for fire activity in the year highlighted by vertical dashed lines. On August 10, farmers and merchants coordinated a joint light of fires in which became known as the Fire Day. August 24 and October 24 marked, respectively, the start and end of the Guarantee of Law and Order (GLO), a governmental decree that authorized the use of army forces into fire combat in the Amazon. (**B**) Monthly fire occurrence in 2019 for the biome, along with the averages of extreme drought years (2005, 2010, and 2015–2016) and regular years from 2003 to 2018, and the range of regular years. (**C**) Similar to (B), but for Amazonian states. (**D**) The percentage shared by each state in total monthly fire occurrence in 2019.

### 3.3. Yearly Active Fire Occurrence, Deforestation, and Mean MCWD for Amazonian States

In Figure 3, we provide annual time-series of active fire occurrence, deforestation and mean MCWD for the three states with anomalous patterns in monthly fire occurrence in 2019 (Acre, Amazonas, and Roraima). Time-series for the other states are presented in Supplementary Materials Figure S1. Annual active fire occurrence in 2019 was the highest of the time-series for Roraima, the third highest

for Amazonas, and in a similar magnitude to previous years for Acre. These three states shared similar patterns of abrupt increases in deforestation from 2018 to 2019, something that was only observable for Pará (Supplementary Materials Figure S1). While in most states annual deforestation in 2019 was the highest of the 2010s decade, it was the second highest of the entire time-series for Amazonas, the highest since 2005 for Acre, and the highest of the time-series for Roraima. The almost threefold increase in fire occurrence from 2018 to 2019 in Roraima was followed by an almost fourfold increase in deforestation. Mean MCWD trends shows that only Roraima had a sharp increase in water deficit from 2018 to 2019, with 2019 presenting one of the lowest mean MCWD of the time-series, together with Amazonas, Pará, Amapá, and Mato Grosso. Except for this last, these states also exhibited the lowest mean MCWD of the time-series in 2017. MCWD levels in Acre over the last years remain somewhat constant and greater than most years of the 2000s.

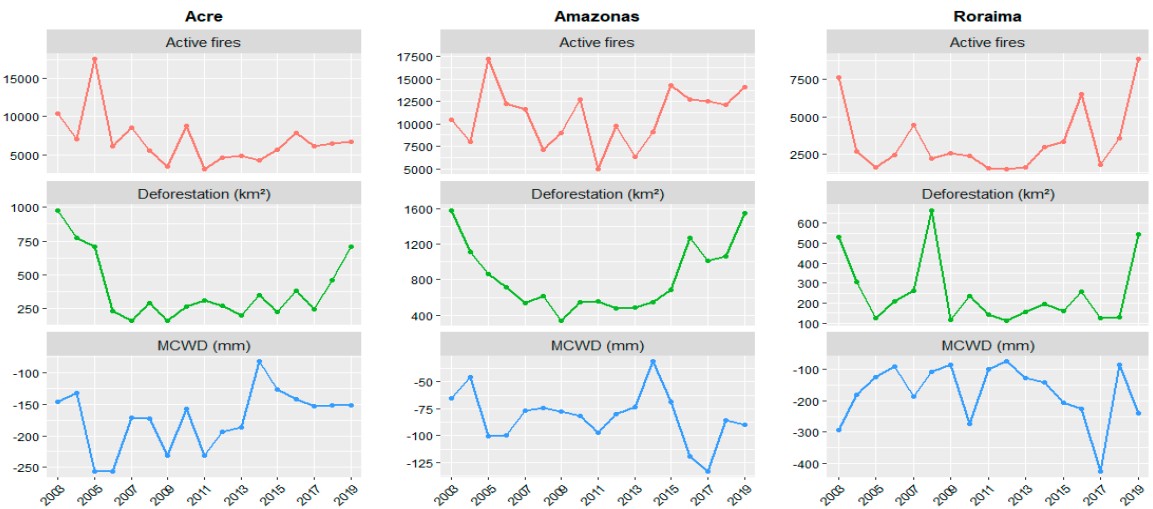

**Figure 3.** Yearly time-series for total number of active fires in red, old-growth deforestation in green, and mean Maximum Cumulative Water Deficit (MCWD) in blue, for three Amazonian states.

### 3.4. Spatial Distribution of Annual Fire Anomalies across the Amazonian Territory and Land Tenure Categories

We identified 1560 across-grid anomalous cells in 2019 (cells with a z-score for fire occurrence two standard deviations above the average across cells), and 1119 cells with maximum fire counts in 2019, which represent respectively 3.62% (156,000 km$^2$) and 2.6% of the Amazon biome area. These two anomaly types share a high degree of spatial overlap, as 46% of across-grid anomalous cells had also a maximum fire count in 2019. Figure 4 shows the distribution of both anomaly types across the biome with respect to states and land tenure categories. Across-grid anomalous cells had a notorious concentration in some particular regions, including the central portion of Roraima state, some clusters in southern Amazonas, eastern Acre, the northernmost portion of Rondônia, southwestern Pará along the federal highway BR-163, among protected areas in southern Pará, and in the agrarian settlements near the Transamazônica highway (BR-230) in northeastern Pará. Among these places, the regions concentrating the highest fire occurrences, as shown by the lowest *p*-values in Figure 4, were central Roraima, southern Amazonas, and northern Rondônia. In terms of states, Pará alone had one-third of the total across-grid anomalous cells (Supplementary Materials Table S1), followed by Mato Grosso (22%) where across-grid anomalous cells were widely distributed. Maximum fire count cells in general were widely dispersed across the territory, but highly clustered in central Roraima and occurring with high density in indigenous lands in central Pará.

Around 43%, 40%, and 28% of the biome's protected areas of sustainable use, strict use, and indigenous lands, respectively, were intersecting any of the two types of anomalous cells. Private lands were present in 89% of across-grid anomalous cells, and agrarian settlements in 40%

(Supplementary Materials Table S2). Undesignated public lands were the second land tenure most present within anomalous cells (57% of both across-grid and maximum fire count), with a high density of intersections in northeastern Rondônia and central Roraima.

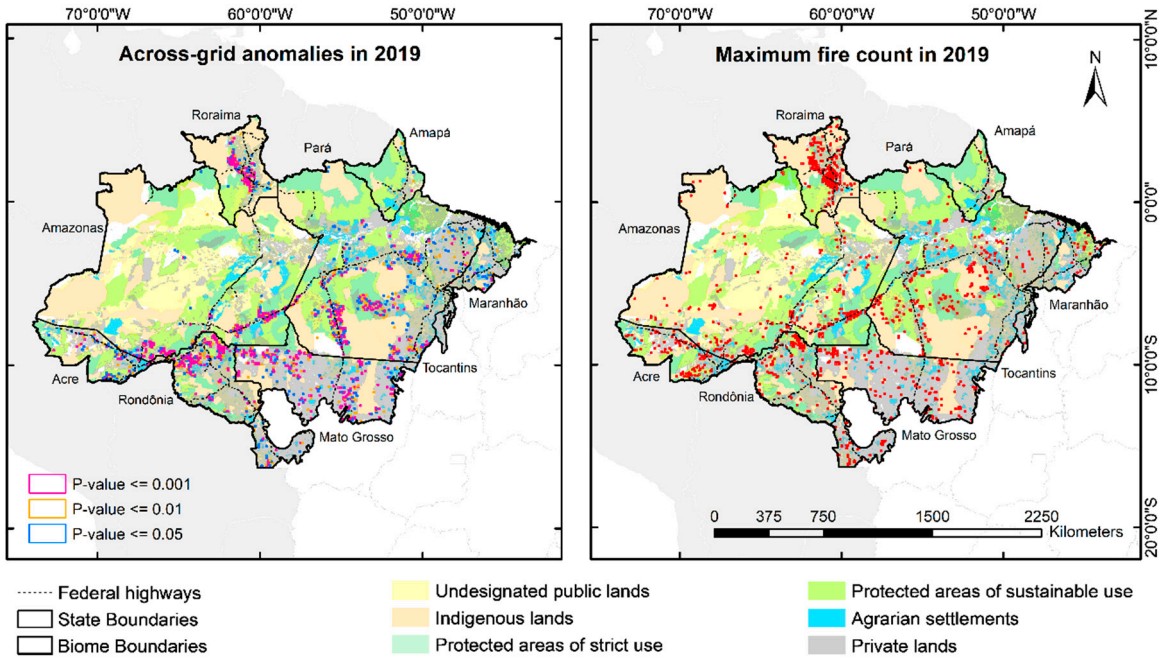

**Figure 4.** Spatial distribution of the two types of fire anomaly across the Brazilian Amazon biome, Amazonian states, and land tenure categories. The first map shows grid-cells with fire occurrence defined as across-grid anomalous (cells with a z-score for fire occurrence two standard deviations above the average across cells), classified according to their *p*-values. The second map shows, in red, the grid-cells in which fire occurrence in 2019 was the maximum of the 2003–2019 period.

### 3.5. Burned Area, Deforestation and Mean MCWD Levels in 2019 across-grid Anomalous Cells

Although across-grid anomalous cells in 2019 covered no more than 4% of the biome's area, they concentrated 47% (21,179 km$^2$) of the total burned area in this year, 64% and 58%, respectively, of the burned area in March and April, and 67% and 52%, respectively, in August and September (Supplementary Materials Table S3). They also concentrated approximately 45% of the total old-growth deforestation in 2019 (Supplementary Materials Figure S2). In fact, 2019 across-grid anomalous cells had their highest annual deforestation of the time-series in 2019 (Supplementary Materials Figure S2). Annual deforestation in 2019 across these cells were two times greater than the average of the time-series and increased by 58% from 2018. In contrast, the mean MCWD in 2019 was 11% lower than the average of the time-series and decreased by 21% from 2018. In a state-specific context, where we compared 2019 values of across-grid anomalous cells in Roraima with those of the entire state (Supplementary Materials Figure S2), anomalous areas also concentrated around 45% of the state deforestation in 2019, and had the highest annual deforestation of the time-series in this year. Moreover, across-grid anomalous cells in Roraima had a sixfold decrease in mean MCWD from 2018 to 2019, compared to a twofold decrease for the state in general.

### 3.6. Recurrence Patterns in Across-Grid Anomalous Cells

Most across-grid fire anomalies in 2019 occurred around the areas with the highest recurrence of fire anomalies in the biome (Figure 5A). Cells with across-grid fire anomalies in 2019 also had across-grid fire anomalies on average in five of the last 17 years. Areas identified as across-grid anomalous for the first time in 2019 can provide us an indication of places where the fire frontier is

expanding. Most places in this sense are concentrated in central Roraima, a region with a low history of intense fire activity. Other regions covering places with a relevant number of first time anomalies include southern Pará and southeastern Amazonas.

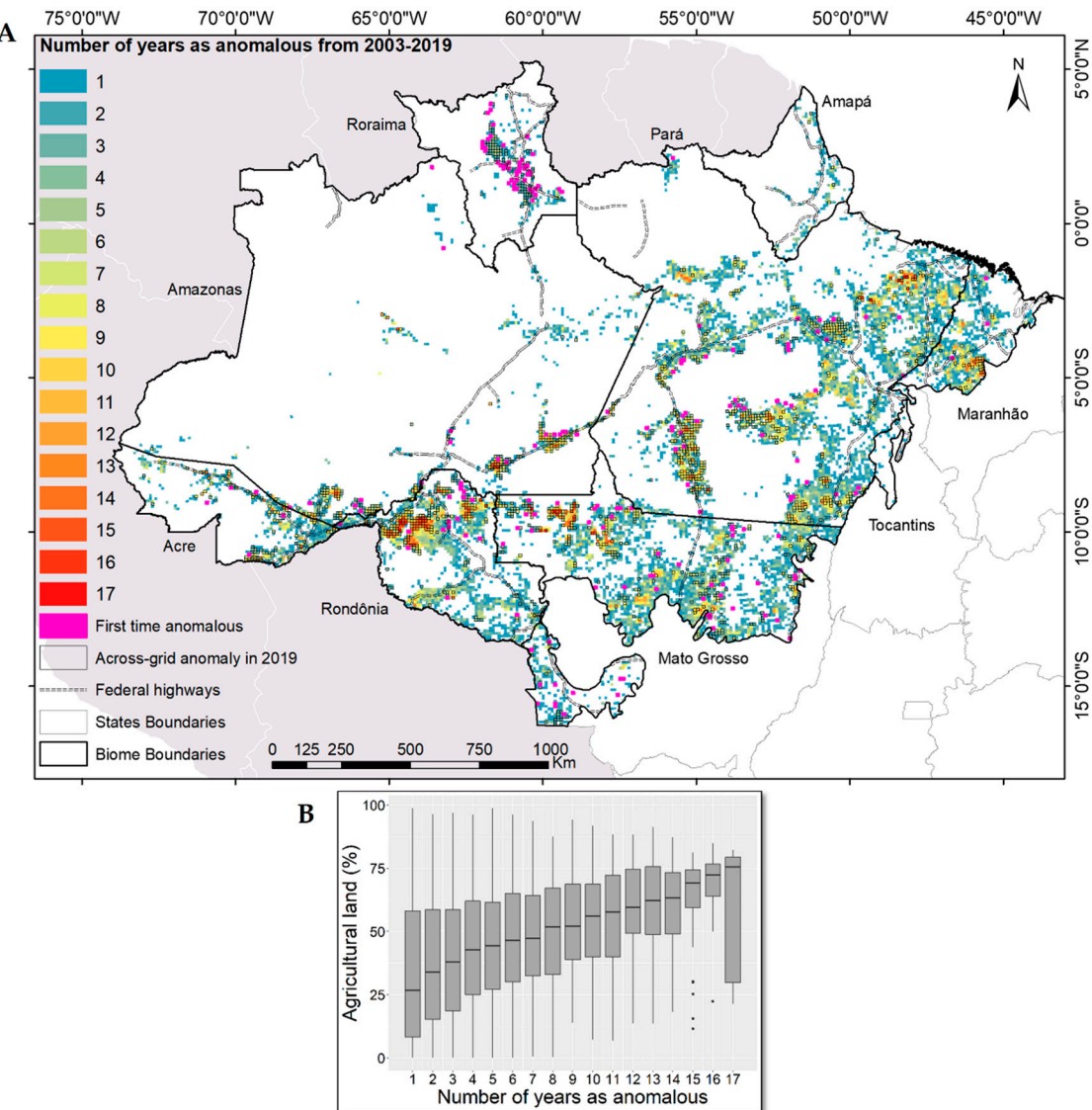

**Figure 5.** (**A**) The number of years each cell in the grid was considered across-grid fire anomalous from the period of 2003–2019. Cells that were across-grid fire anomalous for the first time in 2019 are filled with magenta, while the remaining across-grid fire anomalous cells in 2019 are highlighted in black. (**B**) Boxplots showing the relationship between the number of years a cell was considered across-grid anomalous and the percentage of agricultural land (pasture or cropland) in the cell in 2018. The agricultural area of each cell was calculated over the area of agricultural plus natural vegetation cover (forest, savanna, shrubland, and grassland, except for mangroves).

Historically anomalous areas in terms of fire occurrence are highly widespread across the states of Pará, Mato Grosso, and Rondônia, which have been the greatest contributors to fire activity in the biome. However, recurrence levels seem to follow an ascending gradient towards particular hotspots that have been across-grid anomalous in most years of the last two decades. Such hotspots are mostly present in northern Rondônia, northwestern Mato Grosso, southeastern Amazonas, southern Maranhão, and northeastern and southern Pará. As shown in the boxplots of Figure 5B, as the number of years that a cell has across-grid fire anomalies increases, there is usually an increase in the median percentage

of agricultural cover over natural vegetation cover, though some part of the data did not follow such a pattern. Most areas considered across-grid anomalous for more than 10 years have more than 50% of its area occupied by agriculture. This pattern indicates, for an area under a gradual increase in fire anomaly recurrence (based on a space for time substitution), that every year this area presents an across-grid fire anomaly, there could be a 3% of natural vegetated cover converted to agriculture.

### 3.7. Assessment of the Relationship between Fire Drivers and Active Fire Occurrence in 2019

Figure 6 presents the distribution of total active fires in each grid-cell ($10 \times 10$ km) in 2019 for the 14 variables analyzed. Fire occurrence in 2019 was higher in grid cells with more than 30 fire counts in the three previous years (previous fire), with more than ~3 km$^2$ of deforestation in 2019, and with more than ~4 km$^2$ of deforestation in the five previous years (previous deforestation). Fire occurrence increased as the area of pasture increased up to ~30 km$^2$, and returned to decrease after this threshold, while fire occurrence decreased in areas with more than ~0.4 km$^2$ of croplands. With respect to the three age classes of secondary forests, fire occurrence increased as the area of forests 1–5 years old (Forest Age I) increased up to ~3 km$^2$, but considerably decreased as the area of forests 6–15 years old (Forest Age II) and 16–33 years old (Forest Age III) increased. As for the three fragmentation metrics, fire occurrence increased up to ~350 km$^2$ of total forest edge, up to a mean fragment area of ~2 km$^2$ and up to ~70 fragments per cell, returning to decrease after these thresholds. Fire occurrence was higher in areas with a MCWD between −250 and −200 mm, decreasing with MCWD values lower or greater than this threshold. Fire occurrence slightly increased in areas more than ~7 km away from federal highways (distance to roads) and returned to decrease after ~50 km of distance. Fire occurrence was higher in areas with up to ~10 rural dwellers per cell (rural population density), with no overall tendency as the rural population increased after this threshold.

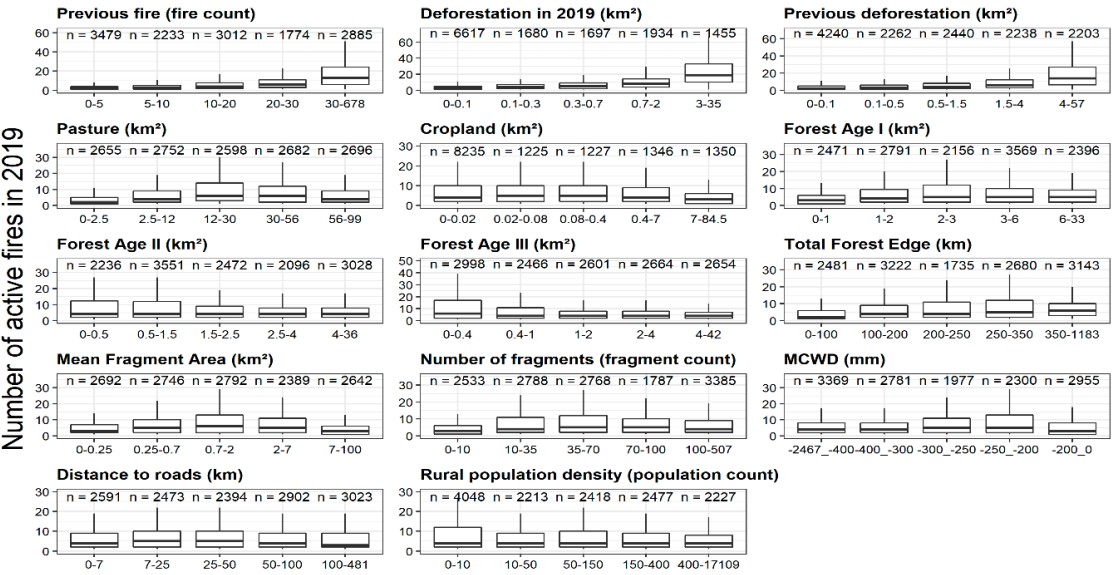

**Figure 6.** Boxplots showing the distribution of 2019 active fires in grid cells according to magnitude intervals in each driver distribution. Each grid cell has an area of $10 \times 10$ km. The "n" on top of each magnitude interval refers to the respective number of grid cells. Outliers were removed from the analysis for visualization purposes.

### 3.8. Testing the Drivers' Power to Discriminate Across-Grid Fire Anomalies at the Biome and State Level

Previous fire occurrence was the variable with the highest power for discriminating across-grid anomalous cells at the biome-scale and for every state, followed by deforestation in 2019 and previous deforestation (Table 2). The states with the greatest area under the curve (AUC) for these three variables

were Acre (AC), Amazonas (AM), and Rondônia (RO). Welch's t-test results for these three variables indicate a greater mean value in cells with across-grid anomalous fire occurrence, compared to cells with non-anomalous fire occurrence. In fact, across-grid anomalous cells in 2019, for the biome in general, had on average around three times more previous active fires, six times more deforestation in 2019 and five times more deforestation in the last five years (Supplementary Materials Table S4).

**Table 2.** The area under the curve (AUC) for each driver, with respect to their ability to discriminate cells with across-grid anomalous fire occurrence from cells with non-anomalous fire occurrence in 2019, for the biome in general and for Amazonian states. AC refer to Acre, AP to Amapá, AM to Amazonas, MA to Maranhão, MT to Mato Grosso, PA to Pará, RO to Rondônia, and RR to Roraima. Values filled in red indicate that anomalous cells had a greater mean than non-anomalous cells, based on Welch's t-test, at the 5% significance level. Values filled in blue indicate a significant lower mean for anomalous cells, and no significant difference was found for values in blank.

| Variable | Area under the Curve (AUC) | | | | | | | | | Mean |
|---|---|---|---|---|---|---|---|---|---|---|
| | | Amazonian States | | | | | | | | |
| | Biome | AC | AP | AM | MA | MT | PA | RO | RR | |
| Previous fire (2016–2018) | 0.83 | 0.92 | 0.9 | 0.92 | 0.85 | 0.78 | 0.82 | 0.88 | 0.83 | 0.86 |
| Deforestation in 2019 | 0.8 | 0.91 | 0.57 | 0.94 | 0.69 | 0.74 | 0.76 | 0.88 | 0.75 | 0.78 |
| Previous deforestation (2014–2018) | 0.8 | 0.87 | 0.52 | 0.94 | 0.69 | 0.74 | 0.77 | 0.85 | 0.76 | 0.77 |
| Pasture | 0.56 | 0.66 | 0.57 | 0.85 | 0.5 | 0.5 | 0.52 | 0.53 | 0.75 | 0.6 |
| Cropland | 0.53 | 0.5 | 0.53 | 0.53 | 0.58 | 0.65 | 0.5 | 0.58 | 0.59 | 0.55 |
| Forest Age I (1–5 years old) | 0.5 | 0.65 | 0.73 | 0.77 | 0.7 | 0.52 | 0.54 | 0.58 | 0.68 | 0.63 |
| Forest Age II (6–15 years old) | 0.65 | 0.51 | 0.72 | 0.63 | 0.56 | 0.62 | 0.69 | 0.67 | 0.55 | 0.62 |
| Forest Age III (16–33 years old) | 0.69 | 0.59 | 0.68 | 0.56 | 0.57 | 0.64 | 0.7 | 0.71 | 0.53 | 0.63 |
| Total forest edge | 0.55 | 0.74 | 0.75 | 0.64 | 0.67 | 0.57 | 0.5 | 0.61 | 0.73 | 0.64 |
| Mean fragment area | 0.51 | 0.64 | 0.7 | 0.73 | 0.5 | 0.54 | 0.52 | 0.56 | 0.65 | 0.59 |
| Number of fragments | 0.52 | 0.64 | 0.74 | 0.73 | 0.57 | 0.52 | 0.5 | 0.52 | 0.54 | 0.59 |
| MCWD | 0.52 | 0.62 | 0.79 | 0.73 | 0.72 | 0.58 | 0.53 | 0.55 | 0.62 | 0.63 |
| Distance to roads | 0.51 | 0.68 | 0.86 | 0.74 | 0.59 | 0.63 | 0.5 | 0.61 | 0.53 | 0.63 |
| Rural population density | 0.62 | 0.58 | 0.69 | 0.6 | 0.58 | 0.58 | 0.65 | 0.65 | 0.52 | 0.61 |

Pasture and cropland were two of the variables with the lowest discriminatory power, with a greater mean for pasture and a lower mean for cropland in across-grid anomalous cells. High AUC values occurred for pasture only in Amazonas and Roraima (RR). Among the three age classes of secondary forest, Forest Age I was the least important and Forest Age III the most at the biome level, but both presented a similar importance at the state level, superior to Forest Age II. Across-grid anomalous cells had a lower mean for Forest Age I at the biome level, but a greater mean at the state level, while mean values for Forest Age II and Forest Age III were always lower in across-grid anomalous cells when significant.

Among the three landscape metrics, total forest edge was the most important and the fourth most important variable from the 14 variables analyzed. The other two landscape metrics presented high AUC values only in AM and AP. Although not important at the biome level, MCWD presented high AUC values in AM, AP, and Maranhão (MA), with a higher mean water deficit in anomalous cells for these states, compared to non-anomalous cells.

Distance to roads was also not important at the biome level, but had high AUC values in AM and AP. Across-grid anomalous cells were on average farther to federal highways in AP, and closer in AM, compared to non-anomalous cells. Across-grid anomalous cells were also closer to federal highways in Acre, and farther in Rondônia, Pará and Mato Grosso. Rural population density was the seventh most important variable at the biome level, but presented low discriminatory power at the state level, with generally a smaller mean for anomalous cells.

*3.9. Relationship of Fire Drivers with Fire Occurrence at the Local Level*

Out of the 14 variables, only the following nine had local relationships with 2019 fire occurrence: previous fire, deforestation in 2019, previous deforestation, pasture, the first two secondary forest-age

classes, and the three fragmentation metrics. A map displaying the spatial distribution in the proportion of fire occurrence variation that is explained by each variable is available in Supplementary Materials Figure S3. Previous fire, deforestation in 2019, and previous deforestation were, respectively, the variables with the greatest number of significant local relationships, presenting a wide distribution across the territory, but especially in Amazonas, Acre, Pará, northern Rondônia, and northwestern Mato Grosso. Other variables with a relevant number of significant local relationships were pasture, total forest edge, and forest age I, which were more prominent in places such as western Acre, northwestern Mato Grosso, southeastern Amazonas, and southwestern Pará.

Considering that previous fire, deforestation in 2019, and previous deforestation were the three most predominant variables, we produced a unique map displaying the "zones of influence" of each driver and their combinations into fire occurrence in 2019 (Figure 7). That is, places where the number of active fires in 2019 increased linearly in a ~ 4000 km$^2$ neighborhood area along with increases in the drivers' value. Local relationships of 2019 fire occurrence with previous fire alone were more predominant in northeastern Pará, Amapá, and Maranhão. Relationships with deforestation in 2019 alone had a predominance in central Rondônia, southeastern Amazonas, and central Pará, while previous deforestation alone was more predominant in western Amazonas, northern Pará, and northwestern Mato Grosso. Among the pairwise combinations, previous fire + previous deforestation had the greatest number of local relationships, remarkably present in northwestern Rondônia, northwestern Mato Grosso, and western Amazonas. Only in a few places 2019 fire occurrence had significant local relationships with all three drivers, most remarkably in northeastern Rondônia, but also in central Acre and some spots in Pará.

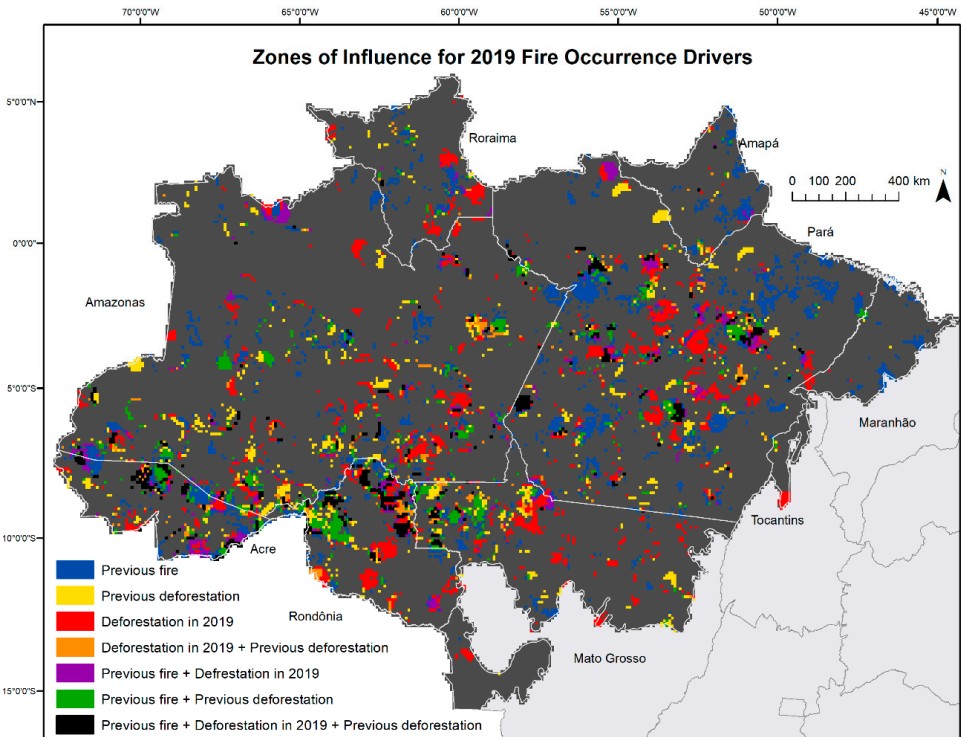

**Figure 7.** Each 10 km pixel in this map represents a statistically significant relationship of a given driver and fire occurrence in 2019 within a region comprising the pixel and its 40 nearest neighbors. In these local relationships, drivers had a positive linear relationship with 2019 fire occurrence and were able to explain at least 50% of fire occurrence variation. The three individual drivers are represented by the primary colors, and their pairwise combinations by the proper secondary colors. Places where all drivers had a significant relationship with fire occurrence are represented in black.

### 3.10. Proximity Patterns in Active Fires and Deforestation

Figure 8A shows the proportion of active fires in a given year within distance intervals of deforested areas in that same year. The proportion of active fires up to 250 m from deforested areas, which are the fires most likely to have occurred within or at the borders of deforested areas, represented around one-fourth of the total for the first years of the time-series. Proportions within this first distance interval were then greatly reduced over the subsequent years, but returned to present sharp increases since 2018. In fact, 2019 had the greatest proportion of fires up to 250 m from same-year deforested areas since 2005, and the greatest proportions within 0.25–0.5 km and 0.5–1 km distances since 2008. On average over the time-series, around one-third of yearly active fires were up to 1 km from deforested areas in the same year. Moreover, there was a constant pattern over the years of at least 50% of yearly active fires being up to 5 km from deforested areas in the same year; this proportion reached 74% in 2019.

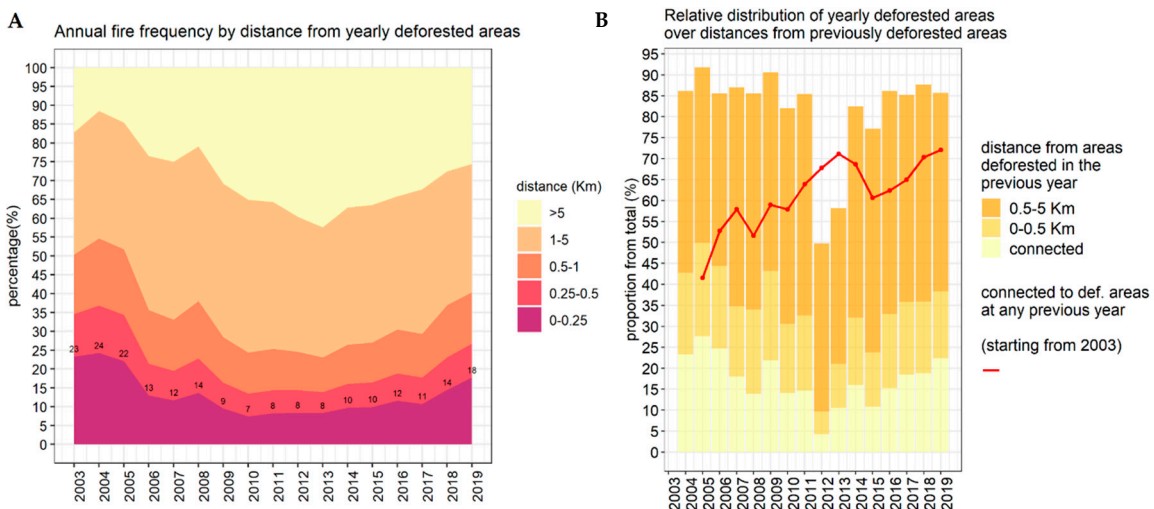

**Figure 8.** (**A**) The proportion of total active fires in a given year within five distance intervals from deforested areas in that same year. (**B**) The proportion of total deforestation polygons in a given year, within specified distances from deforestation polygons in the respective previous year, or connected to deforestation polygons at any previous year, from 2003.

In 2019, the degree of connection between deforested areas in a given year and the previous was the greatest since 2006, with 22% of deforested areas in 2019 connected to deforested areas in 2018 (Figure 8B). Even in years with the lowest proportions of deforestation polygons connected to deforestation polygons in the previous year, such as 2012 and 2013, there was still a high degree of connection to deforested areas at any previous year of the time-series. In 2018 and 2019, around 70% of deforested areas in these years were connected to areas deforested at any period since 2003. On average over the time-series, around one-third and 82% of deforested areas in a given year were, respectively, up to 500 m and up to 5 km from deforested areas in the previous year. The proportion in these two distance intervals for 2019 was, respectively, 38% and 86%.

## 4. Discussion

### 4.1. Were Water Deficit Levels in 2019 Favorable to an Anomalous Fire Activity in Amazonia?

Our results for the Maximum Cumulative Water Deficit (MCWD) yearly variation suggest that 2019 had one of the greatest extent of water deficit since 2003 among years with no extreme droughts. An aggravation of water deficit levels in 2019, however, seemed to be localized. Roraima and Pará were the only two states with a relevant increase in mean water deficit from 2018 to 2019, and in addition to Amapá, Amazonas and Mato Grosso, were the ones in fact having one of the highest mean

water deficit of their time-series in 2019. Nevertheless, abnormally high monthly fire occurrences were only observed for Roraima and Amazonas, while Amapá and Pará had one of their lowest fire occurrences over the last two decades in 2019. In addition, in our driver inspection analyses we found that MCWD was a poor discriminant of 2019 across-grid fire anomalies and did not present significant local spatial relationships with fire occurrence. Areas with across-grid fire anomalies in 2019 had a mean water deficit in this year close to the average of their time-series, and only 20% greater than in 2018, while 2019 deforestation in these areas was the highest of their time-series and 58% greater than in 2018. Our results therefore reinforce that even if an aggravation of water stress can increase forest flammability, ignitions in Amazonia will strictly depend on the occurrence and extent of deforestation and agricultural activities [2]. Findings from Kelley et al. [49] suggest that there is only an 8% probability that burned area levels in August 2019 were linked to 2019 meteorological conditions alone in Acre and southern Amazonas, and a 1% probability in northern Mato Grosso.

Aragão et al. [42] assessed seasonal variations in the Atlantic Multidecadal Oscillation (AMO) index and the Multivariate El Niño Index (MEI), which are robust indicators of droughts in the Amazon, and found index levels greater than the average of the last two decades in all months of 2019 for MEI. Differences in MEI 2019 were more contrasting in February and March, a period corresponding to the beginning of fire season in Roraima. As suggested by Aragão et al. [14], the effects of MEI are more predominant in Roraima, northeastern Amazonas, Amapá, southeastern Mato Grosso, and most of Pará. Such states that would have been mostly affected by the greater-than-average MEI in 2019 conform to those we found that had one of their highest levels of water deficit in 2019. This likely role played by 2019 El Niño into water deficit in these states could have summed with their still incomplete recovery from the 2015 and 2016 extreme droughts. MEI levels in the first semester of 2015 and in the second semester of 2016 were one of the highest of the last 40 years [42], which culminated into the lowest mean MCWD (highest water stress) we observed for these states in 2017 across their time-series, except for Mato Grosso. Findings from Anderson et al. [38] reinforce this idea, as the authors found the greatest anomalies of the 2015/2016 droughts in northern Amazon, compared to the mean of 1981–2016. Furthermore, results from Saatchi [50] support the idea of a prolonged effect of droughts in Amazonian forests, as the authors found that canopy structure signals took four years to recover after the 2005 drought in southwestern Amazonia.

### 4.2. Why the Fire Problem in the Amazon Was So Evident in 2019?

In August 2019, active fire occurrence was greater than the average of non-drought years, but still below the average of drought years at the biome-scale. At the state-level, however, abnormally high fire occurrences in this month were notorious in Acre and Amazonas. Fire occurrence in August 2019 was greater than any non-drought year of the time-series for Acre, and the greatest of the time-series for Amazonas. Nonetheless, while headlines were centered in August, the highest fire occurrence of the last 17 years in March and April for Roraima passed unnoticed.

Our results point to a strong link between deforestation and the anomalous fire activity in these three states, as they also remarkably differed from other states by having spikes in deforestation from 2018 to 2019 and by having one of the highest levels of annual deforestation of their time-series in 2019. Such a link can have a lasting effect on fire trends because it means in practice that more areas are being converted to agriculture, and these areas in turn will be seasonally burned and lead to a greater risk of fire leakage into the fragmented forests. As a result, an intensification of land use change in these states can push their fire trends from what was once anomalous to a new normal over the years, putting fire combat in constraint if funding and hire of personnel do not increase proportionally. Our results indicate that the greater the history of fire anomalies in an area, usually the smaller the proportion of remaining natural vegetation.

The proportion of active fires likely within or near same-year deforested areas has been sharply increasing since 2018 and in 2019 reached the highest level since 2005. This represents an outstanding regress in terms of policy effectiveness, as 2005 marked the year when the Action Plan for the Prevention and Control of Deforestation in the Legal Amazon (PPCDAm) began to take effect. In 2017, Brazil's Government authorized the regularization of undesignated public areas with up to 2500 ha that were illegally occupied up to the year of 2008 (Law nº 13465/2017) [51]. Such a measure could have influenced the sharp increase in deforestation fires we observed since 2018, as it encourages settlers to continue deforesting and converting public lands in hope for similar retributions in the future. Alencar et al. [52] found, for instance, that undesignated public areas had the highest proportion of deforestation in 2019 from land-tenure categories, surpassing private lands (29% and 23%, respectively). We in addition found that undesignated public lands were present in 57% of fire anomaly cells in 2019.

The decision of farmers and other local agents to orchestrate a joint light of fires in the Fire Day also put into evidence this state of weakened environmental governance that has been settling lately in Amazonia. While a response of authorities to Fire Day was slowed by disbelief and lack of inter-institutional cooperation, the event promoted a boost in fire activity in the days following it, which had the highest fire occurrences of the year. The two weeks that followed Fire Day concentrated 20% of fire occurrence over the entire year. If the government had not strengthened fire combat with the Guarantee of Law and Order (GLO) on August 24, this trend would have probably continued in similar magnitudes in September. Conversely, fire activity declined in September and was well below the average of the time-series, while October had the lowest fire occurrence of the last 17 years, demonstrating the critical role played by governance into Amazonian fire crises. Still, 43% of the biome's protected areas of sustainable use, 40% of protected areas of strict use and 28% of indigenous lands were intersecting grid-cells with fire anomalies in this year.

### 4.3. Insights on Fire Occurrence Drivers in Amazonia

Our results brought important contributions to the understanding of fire occurrence drivers in Amazonia. Out of 14 potential drivers, fire occurrence in the three previous years (previous fire), deforestation in 2019, and deforestation in the five previous years (previous deforestation) were the three most related to fire occurrence in 2019. Areas with anomalous fire occurrence better differed from non-anomalous areas by having a greater magnitude of these top three drivers, especially in Acre and Amazonas, reinforcing the idea of an intensification of land use change in these states. We measured other variables related to the degree of anthropic occupation in an area, but our results stress that they will be usually irrelevant to fire anomalies if there are not ongoing processes of intense land use change. Although the top three divers contributed somewhat similarly to fire occurrence, there was a local predominance of either one or two over different regions of the biome, and accounting for this spatial variability could provide for example a more efficient distribution of resources from fire policies.

Previous fire is a variable that indicates the magnitude in which fire lately occurred in an area, which could be for agricultural and deforestation purposes, and due to uncontrolled forest fires. The high performance of this variable points to a tendency of fire anomalies to continue occurring over time in the same areas with a high frequency of active fires. We showed for instance that most areas with anomalous fire occurrence in 2019 were amongst or near clusters with the greatest recurrence of fire anomalies in the biome. We assume that the extent of forest fires may play a great contribution to the number of active fires necessary for an area to be fire anomalous. In this sense, the increasing flammability of forests that had been previously exposed to fires in these areas can contribute to the repeatability of fire anomalies. Cochrane et al. [8] found for instance that not only previously burned forests were more likely to burn than unburned forests, but fire intensity and duration increased with the number of times a forest had been burned.

A strong relationship between annual deforestation and annual fires was also pointed by other studies [53,54], but here we add that deforestation in the five previous years can be contributing as much as annual deforestation to fire occurrence. This variable encompasses several factors that could aid to its high performance. It can indicate for instance burning of recently old-growth deforested areas that were not burned in the previous year due to climatic constraints or legislative deterrence, fire usage in recently converted agricultural areas, and burning of recent vegetation regrowth. We also specifically evaluated three different age classes of secondary forests and found the youngest (1–5 years old) as the most related to fire occurrence at the local level. In addition to the structure and composition of this forest stage that makes it more flammable, studies have shown that this age class is more sought for slash and burn and used as a buffer to other intentional fires [3,55].

The more intense processes of land use change to which areas with fire anomalies are usually subjected reflects directly into the greater fragmentation patterns we observed in these areas, compared to areas with non-anomalous fire occurrence. In a relevant part of the biome, we also found local increases in fire occurrence with increases in forest edge, conforming with the demonstration of many studies that forest fires in the Amazon are more intense and mostly concentrated in the first kilometer from forest edge [6,7,56]. Thus, the likelihood of fire anomalies will partly depend on the spatial extent to which forest edges are exposed to deforestation or agriculture fires. In conditions of high-water stress, such as that for some states in 2019, this likelihood greatly increases in moderate to heavily fragmented landscapes because of the higher penetration of canopy desiccation into the forest interior favoring wildfires [57]. Analyzing fire scars in some Amazonian municipalities, Alencar et al. [58] found that fires penetrated four times further into forests during an El Niño year and burned forest area was 13 times greater in comparison to a non-drought year.

Agriculture in our analyses was disentangled between croplands and pasturelands and we found the latter as much more of a concern to fire activity, at least in the scale of our assessments. Although we acknowledge that the low presence and distribution of croplands in the biome (~56,283 against 525,694 km$^2$ of pasture) could have constrained the proper influence of this variable in our assessments, on the other hand it reinforces that large-scale pasture fires should be treated with more importance in policies given its predominance. In the state of Mato Grosso, more specifically, where the area difference between pasture and cropland is less contrasting than for the biome in general, Cano-Crespo et al. [59] still found a far greater quantity of burned forests edging pasture than edging croplands. In a more site-specific context using field surveys with smallholders, Sorrensen [3] observed that pastures were burned every two to three years, while fallow was burned in a significantly smaller proportion and rotational basis every 15 years.

### 4.4. Lessons Learned to Improve Fire Prevention and Fire Combat Initiatives in the Amazon

Fire prevention and combat can be a daunting task under the vast and heterogeneous territory of the Amazon, demanding for strategic planning for prioritizing resources to the most critical areas. Federal fire institutions in Brazil focus on protected areas, indigenous lands and agrarian settlements, and the remaining areas are kept under the responsibility of state and municipality institutions [60]. There is no public information available on strategies to fire brigade positioning from municipal and state institutions, and the limited information available about federal brigades indicates that they select the top municipalities with the greater number of active fires in the four previous years [23]. The low accuracy that this strategy yields can be partly explained by the fact that the magnitude of fire counts can be either buffered or overestimated by differences in municipality sizes, because in this study we indeed demonstrated that previous fire activity is strongly linked to current fire anomalies, though other drivers play a role as great as.

Stratifying the biome into 10 × 10 km grid cells, our results suggest that the problem is much more local-specific than it seems. Every state has its own particular local clusters where fire anomalies are usually concentrating over time. Moreover, fire anomalies in 2019 were present in no more than 4% of the biome's territory but concentrated almost half of the total burned area in this year, and more

than half of burned area in peak fire season months. Our results point that a strong mechanism to identify these areas in advance is looking at current and recent rates of deforestation, as fire anomalous areas also concentrated 45% of total old-growth forest deforestation for the biome in 2019, and had the highest annual deforestation in their limits in 2019, compared to the last 17 years.

A pattern that fire policies must take into account is that around one-third of yearly active fire occurrence from 2003 to 2019 were up to 1 km from deforested areas in the same year. Such statistic can be accounting not only for deforestation fires, but forest fires that concentrate in the first kilometer from forest edges. The fact that more than 50% of yearly active fires from 2003 to 2019 occurred up to 5 km from yearly deforestation also suggests that the same regions with intense deforestation may also be heavily using fire in agriculture. Therefore, the protected areas and indigenous lands on the fringe of areas with intense deforestation activities are those that will demand more attention from fire brigades. Currently there are daily deforestation alerts in Amazonia being publicly provided by the DETER project [61], and fire combat institutions can use this data to plan ahead of fire season, which is when most clearings will be burned. In fact, planning can begin with at least one year in advance if we consider that around one-third of yearly deforested areas from 2003 to 2019 were up to 500 m from areas deforested in the previous year and most were a continuation of areas deforested in other recent previous years, as our results demonstrated.

The spatial variation in fire season across the biome and the different temporal contributions of each state should also be carefully addressed in fire combat initiatives. Roraima, Amapá and Maranhão for instance require different time-schedules for fire combat planning, as the peak in fire activity do not fit in the general August–September months for these states. Knowing that fire occurrence in Pará usually returns to increase in November, in such a proportion even higher than that of the entire year for some states, the GLO in 2019 could have exceptionally extended until the end of November in this state, as fire occurrence burst again as soon as the GLO ended in October.

We acknowledge that variations in fire season exist even within states, as demonstrated by Schroeder et al. [62], for 2003–2006, and future research could provide an updated and fine scale information of such patterns to help fire combat operations. As our study focused more on the spatial relationship between fire drivers and fire occurrence, further research could provide a complementary perspective by assessing how fire occurrence changes over time with changes in the magnitude of fire drivers. The quantification of these temporal relationships can help estimating the effect of public policies designed to reduce forest wildfire occurrence and guiding future scenarios of carbon emissions in the region.

## 5. Conclusions

The year of 2019 in Amazonia was marked by abnormal increases in monthly active fire occurrence in the states of Roraima, Amazonas, and Acre, which also reached extreme levels of annual deforestation in this year. Roraima most strikingly had the highest active fire occurrence record of the last 17 years in 2019, and a harsh drought condition seemed to have contributed to this achievement in addition to deforestation. Our findings warn of an intensification of land use change processes beyond the infamous Arc of Deforestation, which encompassed the eastern flank of the biome, to the most western portion of the territory. As these states still preserve large swaths of old-growth forests, there is still time and enough information for authorities to provide alternatives to the process of occupation in these areas that are better aligned to a sustainable development agenda and to complying with the Federal Laws. Otherwise, positive anomalies in fire trends could be pushed to a new normal in the future and place forest areas that have long been safeguarded under increasing fragmentation and risk of forest fires. Undesignated public forests are the ones most threatened in this sense, which will require their proper designation to avoid unrestrained deforestation from land grabbing and land speculation.

Fire occurrence in the three previous years, deforestation in 2019 and deforestation in the five previous years were the variables most related to fire occurrence in 2019 among the several we evaluated, regardless of scale level. Fire combat initiatives can greatly improve strategies to brigade allocation by carefully addressing these three variables on a stratification of the territory in samples of 10 km or smaller. Planning for fire combat can start months ahead of fire season by taking into account the proximity patterns we uncovered such as the following: Around one-third of fire occurrences in a given year, from 2003 to 2019, were up to 1 km from deforested areas in the same year; and around one-third of deforested areas in a given year were up to 500 km from deforested areas in the previous year. No wonder that areas with fire anomalies in 2019 concentrated 45% of the total old-growth deforestation in the biome in this year. We stress that fire prevention goes directly in line with strengthening command and control to combat illegal deforestation, and this could avoid a great part of fire crises in Amazonia in years with no extreme droughts. Unfortunately, the increasing trends in deforestation in Brazil, in the opposite direction of goals signed by laws, points that the strong pressure from society to downturn the 2019 fire crisis will continue to be critically necessary if changes are still to come.

**Supplementary Materials:** The following are available online at http://www.mdpi.com/2073-445X/9/12/516/s1. Figure S1: Annual time-series of active fire occurrence, old-growth deforestation and mean Maximum Cumulative Water Deficit (MCWD) for Amazonian states. Table S1: Distribution of fire anomalous cells in 2019 across Amazonian states. Table S2: Proportion of fire anomalous cells in 2019 intersecting each respective land tenure category. Table S3: Proportion of monthly and total burned area in 2019 in the biome within across-grid anomalous cells in 2019. Figure S2: Time-series of active fire occurrence, old-growth deforestation, and mean Maximum Cumulative Water Deficit (MCWD) for the biome, total across-grid anomalous cells in 2019, across grid anomalous cells in Roraima, and total cells in Roraima. Table S4: Welch's t-test results for differences between across-grid anomalous cells in 2019 and cells with non-anomalous fire occurrence, for the biome in general. Figure S3: Local bivariate relationships between each driver and fire occurrence in 2019.

**Author Contributions:** Conceptualization, M.V.F.S., C.A.P., I.S.B., G.O.C., M.S.M., C.C.S.S.L., E.M.M.F., C.G.V.A., A.L.R.F., A.Z.V.M., L.M.E.C., L.O.A., and L.E.O.C.A.; study design, M.V.F.S.; data collection and data preprocessing, M.V.F.S., C.A.P., I.S.B., G.O.C., M.S.M., C.C.S.S.L., E.M.M.F., C.G.V.A., A.L.R.F., A.Z.V.M., and C.H.L.S.J.; formal analysis, M.V.F.S., C.A.P., I.S.B., G.O.C., and C.G.V.A.; writing—original draft preparation, M.V.F.S., writing—review and editing, I.S.B., C.H.L.S.J., L.O.A., and L.E.O.C.A.; supervision, L.O.A. and L.E.O.C.A.; project coordination, M.V.F.S., C.A.P., I.S.B. All authors have read and agreed to the published version of the manuscript.

**Funding:** This study was financed in part by the Coordenação de Aperfeiçoamento de Pessoal de Nível Superior—Brasil (CAPES)—Finance Code 001, by the National Council of Technological and Scientific Development—CNPq through the project "Monitoramento dos Biomas Brasileiros por Satélite – Construção de Novas Capacidades" process: 444418/2018-0, supported by the National Institute for Space Research (INPE), and by the Minas Gerais Research Funding Foundation—Fapemig. The authors thank the SEM-FLAMA project (CNPq/441949/2018-5), ACRE-QUEIMADAS (CNPq/442650/2018-3) and L.E.O.C.A. productivity grant (305054/2016-3). The authors also thank the São Paulo Research Foundation (FAPESP) (process 19/05440-5 and process 2016/02018-2) and the Inter-American Institute for Global Change Research (IAI) SGP-HW 016. The funders had no role in study design, data collection and analysis, decision to publish, or preparation of the manuscript.

**Acknowledgments:** We thank the National Aeronautics and Space Administration (NASA), the MapBiomas Project, and Brazil's National Institute for Space Research (INPE) for providing the freely available datasets used in this study. We also thank the two anonymous reviewers for their insightful feedback on the manuscript.

**Conflicts of Interest:** The authors declare no conflict of interest.

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
