# Peer review of "Drivers of Fire Anomalies in the Brazilian Amazon: Lessons Learned from the 2019 Fire Crisis"

_land, doi:10.3390/land9120516_

Round 1
Reviewer 1 Report
This article represents a very valuable contribution to the literature of fire occurrence analysis, by providing a detailed analysis of climatic and human dirvers of fire ocurrence, at different temporal and spatial scales, in Brazil fires of 2019.
I reccomend the article for acceptance, once some minor suggestions as provided below have been considered.
Specific comments.
The introduction makes a good job in stressing the relevance of the analysis in the Brazilian context. This analysis can also be important in an international context, by analyzing the role of climatic and antropogenic drivers of fire occurrence, in areas of active deforestation.
I would suggest a paragraph to stress the international importance of the current study , with references to global literature of analysis of human and climatic drivers of fire occurrence.
Methods
VIIRS active fire are available since 2012. Please justify the selecion of MODIS (perhaps longer time period available?).
A daily map of fire risk is produced based on weather data in Brazil. Perhaps future studies could analyze this map, in addition to the considered water stress index.
Results.
For water stress, the analysis covered the % of gid cells with water stress anomalies. Perhaps, in addition to figure 1, it might be interesting to show a map of MCWD index or MCWD anomalies against mean values for august 2019, similar to the fire anomalies map shown in fig. 4, to illustrate the spatial distribution of water stress in 2019. A MCWD map might help to visually enphasize wether spatial drivers of fire anomalies in 2019 might be related to spatial patterns of water stress, or instead human patterns were more relevant in explaining fire activity.
Figure 5 suggests a strong human pattern, with historical fire occurrence in all the analyzed years centered around roads. This could be mentioned in the discussion.
This map makes a very good in highlighting how new deforestation occured in 2019 around previous deforestation, as detailedly analyed in section 3.10.
Discussion and conclusions
I would suggest to mention needs for future research.
Reviewer 2 Report
This is a well written manuscript that addresses a critical topic. The analysis of fires in the Amazon biome at state to local levels provides improved insight to factors that drive increased wildfire and that can guide mitigation and control practices.
My specific comments are in the attached annotated file.
